# Infant rhesus macaque (*Macaca mulatta*) personality and subjective well-being

**Elizabeth A. Simpson**[1]*, **Lauren M. Robinson**[2,3], **Annika Paukner**[4]

**1** Department of Psychology, University of Miami, Coral Gables, Florida, United States of America, **2** Konrad Lorenz Institute of Ethology, University of Veterinary Medicine Vienna, Vienna, Austria, **3** Language Research Center, Georgia State University, Atlanta, Georgia, United States of America, **4** Department of Psychology, Nottingham Trent University, Nottingham, England, United Kingdom

* simpsone@miami.edu

## Abstract

Infant temperament is theorized to lay the foundation for adult personality; however, many questions remain regarding personality in infancy, including the number of dimensions, extent to which they are adult-like, and their relation to other outcomes, such as mental and physical health. Here we tested whether adult-like personality dimensions are already present in infancy in a nonhuman primate species. We measured personality and subjective well-being in 7-month-old rhesus macaques (*N* = 55) using the Hominoid Personality Questionnaire and Subjective Well-Being Questionnaire, both of which were developed for adult primates based on human measures. Multiple human raters, who provided infants with daily care since birth, independently rated each infant. We found high interrater reliability. Results from a parallel analysis and scree plot indicated a five component structure, which, using principal components analysis, we found to be comprised of dimensions relating to Openness (e.g., curiosity, inquisitive, playfulness), Assertiveness (e.g., dominance, bullying, aggressive), Anxiety (e.g., vigilance, fearful), Friendliness (e.g., sociable, affectionate, sympathetic), and Intellect (e.g., organized, not erratic). These components are largely analogous to those in adult macaques, suggesting remarkably stable structural personality components across the lifespan. Infant macaques' subjective well-being positively correlates with Openness and Assertiveness and negatively correlated with Anxiety, similar to findings in adult macaques and other primates. Together, these findings suggest that, in macaques, infant personality dimensions may be conceptually related to adult personality and challenge the view that infant temperament may be disorganized and not as meaningful as adult personality. Further research is necessary to explore the antecedents, predictive validity, and stability of these personality components across situations and with development.

## Introduction

Human and nonhuman primate infants display individual differences in various aspects of their psychology and behavior [1–4]. Individual differences in infancy are often described in

**Data Availability Statement:** All relevant data are within the manuscript and its Supporting Information files.

**Funding:** The animal facility was supported by the Division of Intramural Research, Eunice Kennedy

Shriver National Institute of Child Health and Human Development, National Institutes of Health, USA (www.nichd.nih.gov). EAS was supported by National Science Foundation CAREER Award 1653737 (www.nsf.gov). The funders had no role in study design, data collection and analysis, decision to publish, or preparation of the manuscript.

**Competing interests:** The authors have declared that no competing interests exist.

terms of temperament [2–4]. Temperament—a commonly used term to describe infant personality [5]—refers to biologically based inter-individual differences in behavioral tendencies (e.g., attention, motor behavior, emotions, self-regulation), which constitute stable patterns across contexts and over time [6–8]. A related but distinguishable individual difference in infancy is happiness or subjective well-being, which refers to having high levels of life satisfaction [9], high levels of positive affect, and low levels of negative affect [10]. In humans, infant temperament and well-being are considered the early foundations of adult personality and well-being [11,12]; however, many questions remain about their development. For instance, are adult-like personality characteristics present in infancy? That is, to what extent do individual differences in personality traits and well-being appear early and persist across development?

More is known about personality and well-being in human adults relative to in infants or children [10,13]. Adult personality is most commonly measured as five stable domains or constructs (i.e., the Five-Factor Model or the "Big Five"): Extraversion, Openness, Conscientiousness, Neuroticism, and Agreeableness [14,15]. This Five-Factor Model is generalizable across methods of measurement, gender, age, and culture, with strong test-retest reliability and internal consistency [16,17]. These personality dimensions are heritable [18,19] and largely stable but continue to change with age in adults [20,21]. The five factors are predictive of academic performance [22], career success [23], romantic relationship satisfaction [24], health [25], and subjective well-being [26].

Subjective well-being—a construct typically measured in adults through self-reports—includes individual differences in emotions, such as positive affect and happiness, and cognitive components, such as goal achievement and life satisfaction [27,28]. Subjective well-being is related to, but also distinguishable from, personality [29]. For example, higher levels of Extroversion, Agreeableness, Conscientiousness, and emotional stability (i.e., lower Neuroticism) are associated with greater subjective well-being [13]. Subjective well-being is heritable [30], and, in adults, largely stable over time [31] but also still changing with age [32], and positively associated with mental and physical health [33,34].

## Human infant personality and subjective well-being

Given the role of personality and well-being for predicting health and success, it is important to uncover their early roots and how they emerge and develop. While it is theorized that the adult five-factor personality dimensions are largely synonymous with infant temperament [11,35], to date, there have been few tests of this proposal, and therefore, there is rather limited support for this idea. Toddlers' temperaments predict their five-factor personality scores into later childhood and adolescence [36–39], which suggest some degree of stability in personality. However, little is known about whether these dimensions are present earlier in development, in infancy [36,40,41].

Few studies have examined infant well-being and whether there are stable intra-individual differences. One study found that parent reports of 1-year-old infants' temperament predicted infants' later life satisfaction as adults through 29 years of age [10]. This longitudinal study found that infants' levels of positive affect, in particular, predicted their life satisfaction as adults, whereas infants' negative affect, in contrast, did not predict any measures of adult well-being. Though limited, there is also some evidence of an association between infant personality and later subjective well-being. For example, one study found that the developmental trajectories of externalizing behavior problems (e.g., aggression, temper loss, noncompliance), from infancy (1.5 years old) to mid-adolescence (14 years old), are associated with well-being in young adults, at 18 years of age [42]. However, to our knowledge, there are no studies that

have directly measured both well-being and personality in infancy to explore their development and relation to one another. In sum, we know little about the early emergence of subjective well-being in infancy or how it is related to other dimensions of infant personality. Nor do we understand the extent to which these dimensions may be adult-like, established early in development.

## Value of animal models

Studies in animals may shed light on the early development of individual differences in personality and well-being, which are not unique to humans. Indeed, personality in animals is a well-established phenomenon across a wide range of species, including octopi [43], dogs [44], snakes [45], zebra finches [46], bees [47], and whales [48]. While animal studies of personality are interesting in their own right, they also widen our understanding of human personality through enabling approaches to questions that are difficult or impossible to answer with studies in humans [49–51]. Studies of personality in animals are necessary to clarify the phylogenetic history of specific traits, offering insights into their evolutionary origins [52,53]. Furthermore, animals are useful for developmental studies of personality, as many species exhibit more rapid development and shorter lifespans, making it possible to longitudinally measure personality over the lifespan in a shorter period of time and with less attrition [54].

Finally, in animals, there is more experimental control and manipulability, enabling more accurate measures of prenatal and postnatal contributions to personality and well-being, which would be ethically or practically difficult if not impossible in humans [55–59]. For instance, in humans, infant and parent well-being are linked, with infants' temperaments affecting maternal well-being [60] and maternal well-being, during and after pregnancy, affecting children's well-being [61]. Determining causal relations among these complex systems is challenging. Animal studies can overcome these limitations by enabling a high degree of experimental control over infants' environments. By standardizing infants' environments, for example, this can help disentangle environmental contributions to individual differences in personality or well-being observed in a specific sample. For example, in chimpanzees, infants who were raised in an environment that included less contact with conspecifics, compared to infants reared in more species-typical environments, displayed lower levels of extroversion later in life [62]. At the same time, caution is warranted when generalizing findings from one sample to another, when infants' early environments vary substantially [63].

The Five-Factor Model has been adapted for a variety of nonhuman primate species, although the number and nature of the factors varies somewhat across species [64–68]. These studies often use an approach similar to that used with human infants, assessing personality and well-being through knowledgeable informants, such as animal care staff [69,70], which reveal strong levels of inter-observer agreement, and predictive validity of behaviors in various real-world contexts [49]. For example, rhesus macaques are reported to have six personality components: Confidence, Friendliness, Dominance (hereafter referred to as Assertiveness to avoid confusion with traditional measures of hierarchical dominance), Anxiety, Openness, and Activity [53]. Much like in humans, these personality dimensions are heritable [71,72] and are associated with specific patterns of behavior. For example, individuals rated higher in sociability (Extraversion), tend to engage in more affiliative interactions, whereas individuals higher in confidence tend to engage in more aggressive behaviors [73]. Furthermore, some of these rhesus macaque personality dimensions are associated with lifetime injury incidence [74] and well-being [53].

Subjective well-being has been reported as a valid measure in a variety of nonhuman primate species [30,75–79]. In studies of nonhuman animal subjective well-being, human raters

are asked how often each animal is happy, how satisfied each animal is with their social relationships, how successful each animal is in achieving their goals, and to imagine how happy they would be if they were that animal for a week [80]. Such studies reveal that, in adult macaques, much like in humans and chimpanzees, higher confidence and friendliness, and lower anxiety are associated with higher subjective well-being [53]. However, we know little about the early emergence of subjective well-being in infancy or how it may relate to other dimensions of infant personality. Nor do we understand the extent to which these dimensions may be adult-like, established early in development.

## Infant macaque personality

A barrier to understanding personality development in infancy is the lack of well-established measures. One approach involves placing infants into various situations (e.g., novel environment, person, or object), and measuring their behavioral and physiological reactions (e.g., stress-related behaviors, salivary cortisol). An example of this approach is the Brazelton Newborn Behavioral Assessment Scale (NBAS), which is the most common measure of temperament in human newborns and has also been adapted for macaque newborns (Infant Behavioral Assessment Scale, IBAS; [81]). The IBAS focuses primarily on neurological development, including sensory and motor abilities (e.g., reflexes, orienting). Similarly, the Biobehavioral Assessment (BBA) is designed to assess 3- to 4-month-old macaques' behavioral and physiological responses to a variety of stressors over a 48-hour period [63]. While these approaches offer valuable insight into activity levels, irritability, and stress-related aspects of infants' development, there remains a need to better capture more positive types of traits, such as infants' curiosity, playfulness, and sociability. Both the IBAS and the BBA assessments are also costly and labor-intensive, requiring animal handling by trained staff, as well as intensive behavioral scoring by reliable observers.

An alternative method, which may give us a broader view of personality dimensions, is to use caregiver surveys [62]. In humans, parental surveys capitalize on the fact that caregivers have extensive observations of their infants across a wide variety of contexts and are therefore one of the easiest, most reliable, and most predictive measures of temperament [82,83]. Similarly, animal care technicians, who interact with individual animals daily over the course of many months, and sometimes many years, have been shown to provide animal personality ratings that are consistent across raters and over time [53, 67,77,84–87]. Early surveys of macaque infant personality were developed prior to the human Five-Factor Model, which has expanded our understanding of a wider variety of personality domains [87]. Previous studies, therefore, may not have captured all of the dimensions of infant macaque personality [53].

## Current study

In the current study, we examined personality in infant rhesus macaque monkeys raised in a well-controlled, standardized, laboratory environment by human caretakers. This unique early environment offered a degree of experimental control and standardization that enabled us to observe natural variation in personality with limited environmental influences, a level of control impossible to achieve in human studies of personality and subjective well-being. We explored whether caretakers—who were intricately familiar with each infant through daily interactions since birth—could reliably rate infant macaque personality and subjective well-being. We also examined whether there were personality dimensions in infant monkeys that resemble those in adult monkeys (i.e., six component adult rhesus macaque structure: Table 1 in [53]). Finally, we tested whether infant personality dimensions are related to their well-being.

## Materials and methods

### Ethics statement

This study was purely observational. The *Eunice Kennedy Shriver* National Institute of Child Health and Human Development Animal Care and Use Committee approved all animal procedures. We conducted the study in accordance with the Guide for the Care and Use of Laboratory Animals and complied with the Animal Welfare Act. During the course of this study, infants were fed with Similac® Advance® (Abbott Laboratories) and, starting at 2 weeks old, Purina Monkey chow (#5054). Additional food enrichment, including fruits, seeds, and nuts, was introduced twice daily when infants were 2 months old. Water was available ad libitum. Infants' housing was enriched by an inanimate surrogate mother covered with fleece fabric as well as blankets and various plastic and rubber toys, which were rotated daily. At the conclusion of data collection for the current study, infants continued to be housed in the nursery as part of ongoing, unrelated research studies until ca. 6–8 months of age, after which they were transferred to large peer groups. See [88] for further details on housing, enrichment, and feeding.

### Subjects

We studied 55 healthy, full-term infant rhesus macaques (*Macaca mulatta*), including 29 females and 26 males born in four cohorts between April of 2013 and July of 2016: $N = 18$ (7 females) born in 2013, $N = 10$ (6 females) born in 2014, $N = 16$ (9 females) born in 2015, and $N = 11$ (7 females) born in 2016. Infants were housed in the Animal Care Center at the *Eunice Kennedy Shriver* National Institute of Child Health and Human Development, National Institutes of Health. Infants were rated when they were between 6.5 and 7.5 months old (196 to 225 days old; mean (*SD*) = 211 ± 9 days old). Infants were separated from their mothers on the day of birth (typically by 8am) and reared in a nursery facility by human caretakers for ongoing, unrelated research studies. Infants were housed in adjacent incubators (51 cm × 38 cm × 43 cm) for the first 2 weeks of life and in larger cages (65 cm × 73 cm × 83 cm) thereafter. Human caretakers were present for 13 hours each day and interacted with infants every 2 hours for feeding and cleaning purposes. In both housing arrangements infants could see and hear other infants.

In the first 5 weeks after birth, infants were singly housed and raised identically. Once the youngest infant reached 36 days of age, infants were placed into small, same-aged peer groups. Infants were randomly assigned to one of two rearing conditions for unrelated research studies: low-socialization infants ($N = 27$) and high-socialization infants ($N = 28$). Low-socialization infants continued to be individually housed but assigned to playgroups composed of 3 to 4 peers and put together for 2 hours a day, 5 days a week. High-socialization infants were raised in groups with 3 to 4 peers (for details: [88–90]). By 6 months of age, all infants had extensive experience with same-aged conspecifics. Between 6–8 months old, all infants from each year cohort were placed into one large peer group together with one adult male and several mother-reared infants born the same year (for details see [91]). Therefore, rearing experiences converged after this initial period of differential rearing.

We choose to assess personality and well-being in these infants while they were still in the nursery setting, with one-on-one interactions occurring daily between animal care staff and the infants. At the same time, we waited until infants were 7 months of to give the raters time to get to know their individual personalities.

### Measures

Each infant monkey's personality and subjective well-being was rated by two to three of the full-time animal care staff who worked with the animals. These six raters had observed and

interacted with the infants since the day they were brought to the nursery. Raters were asked to make their judgments on the basis of their own understanding of each trait and the descriptions of each trait that were provided. They were instructed to use the monkey's behaviors and interactions with other monkeys to make their ratings, considering their understanding of typical monkey behavior, to decide if a particular monkey is above, below, or average for each trait. In written and verbal instructions, each rater was instructed to keep their ratings private and not discuss their ratings with the other raters. Infant macaques (*N* = 55) were rated when they reached approximately 7 months of age by at least two raters (mean = 2.8 raters per subject; range 2 to 3 raters per subject). We collected data between November 2013 and February 2017. For the personality ratings, there were no missing items out of 8,316 items ratings; for the subjective well-being ratings there were no missing items out of 616 item ratings.

**Personality.** We measured personality using the Hominoid Personality Questionnaire (HPQ) [92], which is 54-item questionnaire where each item is made up of an adjective and 1–3 descriptive sentences. As an example, the item 'gentle' is presented as, "GENTLE: Subject responds to others in an easy-going, kind, and considerate manner. Subject is not rough or threatening." Each item is followed by a 7-item Likert scale with answers ranging from 1 "Least: Displays either total absence or negligible amounts of the trait" to 7 "Most: Displays extremely large amounts of the trait." The HPQ can be downloaded from [93].

**Subjective well-being.** Each macaque was rated on subjective well-being, a four-item questionnaire based on King and Landau's questionnaire [80]. Using this questionnaire, raters were asked to answer questions on how often each animal is happy, how successful each animal is in achieving their goals, to imagine how happy they would be if they were that animal for a week, and to estimate how satisfied each animal is with their social relationships. Each question is followed by a 7-item Likert scale with answers ranging from "Displays either total absence or negligible amounts of the trait or state" to "Displays extremely large amounts of the trait." The subjective well-being questionnaire can be downloaded from [94].

## Data analysis

All analyses were conducted using R, version 3.5.1 [95]. Principal components analyses and parallel analysis were conducted using the psych package [96]. The R script is available in supporting materials.

**Item interrater reliabilities.** We used two intraclass correlations (ICC) to estimate interrater reliabilities among raters [97]. *ICC*(3,1) measures the reliability of individual ratings whereas *ICC*(3,*k*) measures the reliability of average ratings across *k* raters.

**Principal components analyses.** After the ICCs were performed all data were averaged across raters resulting in a single score for each animal, and these scores were used for all remaining analyses. To examine the structure of infant macaque personality we used a principal component analysis (PCA). We determined the numbers of components to extract by using a parallel analysis and examining the scree plot [98,99] using the fa.parallel function in the 'psych' package [96]. We followed the comparative personality methods outlined by Robinson et al. [100] and discussion by Weiss [101] and calculated two additional structures, a structure with one less component than recommended by the parallel analysis and a structure with one more component. We examined each structure with both varimax and promax rotations of the structure(s); if the correlations in the promax rotation were relatively low (below *r* = 0.40) then we used the varimax rotation.

Following this analysis, we next computed unit-weighted component scores [102], based on the derived structures and the published adult six component structure [53] where a weight of +1 was assigned to loadings that were greater or equal to .4 and a weight of -1 was assigned to

loadings that were equal to or less than -.4; all other loadings were assigned weights of 0. If an item loaded at greater than or equal to |.4| on multiple components, then the item was assigned to the component on which it loaded the highest. We performed a single PCA and followed the statistical procedure to determine the structure derived from the four subjective well-being items.

**Component interrater reliabilities.** To check interrater reliability at the component level, we created unit-weighted component scores [102] of individual scores based on the results of the PCA again using $ICC(3,1)$ and $ICC(3,k)$.

**Pearson's correlations.** We ran two sets of Pearson's correlations. We first tested for associations between the infant macaque personality component scores and the previously published adult macaque personality component scores [53]. This approach allowed us to determine which structure most closely resembled the adult six component structure (Table 1 in [53]). In the second set of correlations, we tested for associations between the infant macaque personality component scores and the infant macaque subjective well-being component scores.

## Results

### Interrater reliabilities

Observers were found to agree on all four subjective well-being items and all but two HPQ items (Table 1). For the subjective well-being items the mean $ICC(3,1)$ was 0.45 ($SD \pm 0.08$, range = 0.34 to 0.52) and the $ICC(3,k)$ was 0.69 ($SD \pm 0.07$, range = 0.59 to 0.75). For the HPQ items the mean $ICC(3,1)$ was 0.32 ($SD \pm 0.17$, range -0.14 to 0.66) and the $ICC(3,k)$ was 0.52 ($SD \pm 0.25$, range -0.52 to 0.85). We excluded the HPQ items for which observers were unreliable—unperceptive and imitative—from further analysis.

### Principal components analyses

The parallel analysis and scree plot of the personality items suggested a five component structure (see Table 2), which accounted for 62% of total variance. The promax rotation (S1 Table) showed relatively low correlations between components (highest correlation = 0.35; S2 Table) therefore we decided to interpret the varimax rotated five component structure. We report the promax rotated four, five, and six component structures in S1 Table; see S2 Table for the component correlations. We report the varimax rotated four and six component structures in S3 Table. We also ran a factor analysis (see S4 Table) and compared the results to this structure using a congruence test. We found the results of both tests to be virtually identical (congruence = 1.00 across all corresponding components) and therefore continued with the PCA approach to be consistent with the method used in Weiss [53]. For the suggested five component structure the mean $ICC(3,1)$ was 0.49 ($SD \pm 0.18$, range 0.21 to 0.64) and $ICC(3,k)$ was 0.71 ($SD \pm 0.17$, range 0.42 to 0.83) (see S5 Table). The parallel analysis of the subjective well-being items suggested a single component structure (see S6 Table).

For the five component varimax rotated infant structure, the first component was comprised of items relating to curiosity, activity, and innovation such as, inquisitive, playful, and inventive; we named this component Openness. The second component was comprised of items relating to dominance traits such as bullying, aggressive, defiant, and manipulative; we named this component Assertiveness. The third component was comprised of items relating to anxiety and vigilance such as fearful, timid, excitable, vulnerable, and anxious; we named this component Anxiety. The fourth component was comprised of items relating to sociability such as affectionate, sympathetic, helpful, and friendly; we named this component Friendliness. The fifth component was comprised of items relating to decision making behavior such

**Table 1. Interrater reliability of subjective well-being and hominoid personality questionnaire items.**

| Subjective Well-Being Item | ICC(3,1) | ICC(3,k) |
|---|---|---|
| Time animal is happy | 0.52 | 0.75 |
| Goal achievement | 0.49 | 0.73 |
| Happiness as animal | 0.47 | 0.71 |
| Social satisfaction | 0.34 | 0.59 |
| Subjective well-being average | 0.45 | 0.69 |
| **Personality Item** | **ICC(3,1)** | **ICC(3,k)** |
| Dominant | 0.66 | 0.85 |
| Timid | 0.62 | 0.82 |
| Submissive | 0.61 | 0.82 |
| Cautious | 0.60 | 0.81 |
| Aggressive | 0.60 | 0.81 |
| Bullying | 0.56 | 0.78 |
| Curious | 0.54 | 0.77 |
| Fearful | 0.49 | 0.73 |
| Playful | 0.44 | 0.69 |
| Inquisitive | 0.44 | 0.69 |
| Helpful | 0.44 | 0.69 |
| Anxious | 0.44 | 0.68 |
| Independent | 0.43 | 0.68 |
| Autistic | 0.42 | 0.67 |
| Active | 0.42 | 0.67 |
| Thoughtless | 0.40 | 0.66 |
| Stingy | 0.40 | 0.65 |
| Reckless | 0.39 | 0.64 |
| Individualistic | 0.39 | 0.64 |
| Persistent | 0.38 | 0.63 |
| Depressed | 0.38 | 0.63 |
| Affectionate | 0.38 | 0.63 |
| Manipulative | 0.37 | 0.63 |
| Cool | 0.36 | 0.61 |
| Stable | 0.36 | 0.61 |
| Impulsive | 0.36 | 0.61 |
| Solitary | 0.33 | 0.58 |
| Jealous | 0.33 | 0.58 |
| Vulnerable | 0.33 | 0.58 |
| Dependent | 0.31 | 0.56 |
| Sociable | 0.31 | 0.56 |
| Protective | 0.29 | 0.54 |
| Gentle | 0.29 | 0.53 |
| Distractible | 0.28 | 0.53 |
| Sympathetic | 0.28 | 0.52 |
| Friendly | 0.27 | 0.51 |
| Inventive | 0.27 | 0.51 |
| Lazy | 0.26 | 0.50 |
| Disorganized | 0.23 | 0.46 |
| Irritable | 0.22 | 0.45 |
| Decisive | 0.21 | 0.42 |

*(Continued)*

**Table 1.** (Continued)

| | | |
|---|---|---|
| Conventional | 0.20 | 0.42 |
| Innovative | 0.18 | 0.38 |
| Defiant | 0.17 | 0.37 |
| Erratic | 0.16 | 0.35 |
| Intelligent | 0.15 | 0.33 |
| Clumsy | 0.14 | 0.32 |
| Excitable | 0.14 | 0.31 |
| Sensitive | 0.07 | 0.18 |
| Unemotional | 0.07 | 0.17 |
| Quitting | 0.05 | 0.13 |
| Predictable | 0.01 | 0.04 |
| Unperceptive | -0.06 | -0.18 |
| Imitative | -0.14 | -0.52 |
| Hominoid Personality Questionnaire average | 0.32 | 0.52 |

Intraclass correlations (ICC) were based on 55 rhesus macaques, the number of raters (*k*) ranged between 2 and 3, *k* = 2.8.

as intelligent, decisive, and predictable; as this dimension appeared to resemble orangutan Intellect (Table 3 in [79]) rather than any one adult rhesus macaque dimension, we named this component Intellect.

The four and six component structures accounted for 57% and 65% of total variance, respectively. These structures resembled that of the five component with the primary exception being the items that loaded onto Intellect. In the four component structure (S3 Table) four of the intellect items (disorganized, excitable, intelligent, erratic) loaded onto the Anxiety component with two items (predictable and clumsy) not loading onto any component. In the six component structure three of the Intellect items (erratic, predictable, and clumsy) loaded onto the fifth component, which we named Predictability. The two remaining Intellect items (decisive and intelligent) loaded onto the sixth component, which we called Intellect. The promax rotated four, five, and six component structures can be found in S1 Table; the varimax rotated four and six component structures can be found in S3 Table.

All four subjective well-being items loaded onto a single component (S4 Table). This structure accounted for 69% of variance. This result matches the structure found in adult rhesus macaques [53].

## Pearson's correlations

We tested for correlations between our infant macaque personality component scores and the component scores based on the published adult macaque personality structure (see Table 3 in [53]). Adult macaque Confidence was significantly correlated ($ps < 0.05$) with higher infant Openness ($rs \geq 0.65$), higher infant Assertiveness ($rs = 0.62$), and lower infant Anxiety ($rs \geq -0.84$) in the infant four, five, and six component structures and with higher infant Intellect ($r = 0.72$) in the infant six component structure (Table 3). Adult macaque Openness was significantly correlated with higher infant Openness ($rs \geq 0.94$) in the four, five, and infant six component structures and with lower infant Intellect ($r = -0.50$) in the infant five component structure and lower infant Predictability in the infant six component structure ($r = -0.47$). Adult macaque Assertiveness was correlated with higher infant Openness ($rs \geq 0.52$) and

**Table 2. Varimax rotated infant rhesus macaque structure.**

| Item | Infant Macaque Components | | | | | | Corresponding Adult Component |
|---|---|---|---|---|---|---|---|
| | Openness | Assertiveness | Anxious* | Friendliness | Intellect* | $h^2$ | |
| Curious | **0.80** | 0.02 | -0.18 | 0.17 | -0.10 | 0.72 | Openness + |
| Active | **0.80** | 0.09 | -0.02 | -0.13 | -0.09 | 0.68 | Activity + |
| Inquisitive | **0.78** | -0.09 | -0.23 | 0.10 | -0.13 | 0.70 | Openness + |
| Lazy | **-0.78** | -0.01 | -0.23 | -0.02 | -0.04 | 0.66 | Activity - |
| Playful | **0.76** | 0.20 | -0.16 | 0.17 | -0.04 | 0.67 | Activity + |
| Impulsive | **0.67** | 0.20 | 0.08 | -0.19 | **-0.41** | 0.70 | Openness + |
| Depressed | **-0.65** | -0.08 | 0.27 | -0.29 | -0.13 | 0.60 | Friendliness - |
| Reckless | **0.63** | 0.39 | -0.20 | -0.02 | -0.37 | 0.73 | Dominant + |
| Distractible | **0.62** | -0.01 | 0.06 | 0.14 | **-0.51** | 0.68 | Confidence - |
| Timid | **-0.62** | -0.26 | **0.50** | -0.10 | 0.18 | 0.74 | Confidence - |
| Innovative | **0.59** | 0.19 | -0.08 | 0.01 | 0.16 | 0.42 | Openness + |
| Inventive | **0.58** | 0.20 | -0.25 | 0.08 | 0.06 | 0.45 | Openness + |
| Thoughtless | **0.57** | 0.00 | -0.02 | 0.16 | **-0.56** | 0.67 | Openness + |
| Cautious | **-0.56** | -0.36 | 0.36 | -0.09 | 0.33 | 0.68 | Confidence - |
| Persistent | **0.54** | **0.51** | -0.13 | -0.16 | -0.04 | 0.60 | Friendliness + |
| Individualistic | **0.47** | 0.22 | 0.21 | -0.32 | -0.21 | 0.47 | Dominant + |
| Sensitive | -0.37 | -0.21 | 0.27 | 0.13 | 0.21 | 0.32 | Friendliness + |
| Aggressive | 0.08 | **0.88** | -0.11 | -0.05 | 0.01 | 0.80 | Dominant + |
| Bullying | 0.09 | **0.87** | -0.16 | -0.03 | 0.02 | 0.79 | Dominant + |
| Dominant | 0.25 | **0.81** | -0.23 | 0.03 | 0.14 | 0.78 | Dominant + |
| Gentle | -0.02 | **-0.75** | -0.17 | 0.27 | 0.08 | 0.66 | Dominant - |
| Defiant | 0.21 | **0.72** | -0.04 | 0.11 | -0.31 | 0.68 | Dominant + |
| Stingy | 0.16 | **0.72** | -0.01 | -0.07 | -0.03 | 0.55 | Dominant + |
| Submissive | -0.20 | **-0.72** | **0.40** | 0.06 | -0.15 | 0.74 | Confidence - |
| Manipulative | 0.00 | **0.67** | -0.09 | 0.20 | 0.11 | 0.51 | Dominant + |
| Jealous | 0.06 | **0.64** | 0.14 | 0.02 | -0.14 | 0.45 | Anxious + |
| Irritable | **-0.48** | **0.59** | 0.17 | -0.20 | -0.22 | 0.69 | Dominant + |
| Quitting | -0.19 | -0.36 | 0.04 | 0.07 | -0.13 | 0.19 | Anxious + |
| Conventional | -0.34 | -0.36 | -0.29 | 0.30 | 0.29 | 0.50 | Activity - |
| Cool | 0.18 | -0.03 | **-0.79** | 0.15 | 0.22 | 0.73 | Anxious - |
| Stable | 0.25 | 0.03 | **-0.77** | 0.19 | 0.09 | 0.70 | Confidence + |
| Unemotional | -0.09 | 0.01 | **-0.71** | 0.01 | -0.04 | 0.51 | Anxious - |
| Anxious | **-0.46** | -0.12 | **0.68** | -0.24 | -0.08 | 0.75 | Anxious + |
| Fearful | **-0.52** | -0.14 | **0.65** | -0.13 | 0.11 | 0.75 | Confidence - |
| Excitable | 0.18 | 0.06 | **0.65** | 0.02 | -0.26 | 0.53 | Dominant + |
| Autistic | -0.13 | -0.07 | **0.60** | -0.18 | -0.15 | 0.43 | N/A |
| Vulnerable | -0.09 | **-0.50** | **0.59** | 0.11 | -0.10 | 0.63 | Confidence - |
| Affectionate | 0.13 | -0.13 | -0.15 | **0.81** | -0.09 | 0.73 | Friendliness + |
| Helpful | 0.21 | -0.14 | -0.36 | **0.71** | -0.13 | 0.72 | Friendliness + |
| Sympathetic | 0.13 | -0.27 | -0.11 | **0.70** | -0.03 | 0.60 | Friendliness + |
| Protective | 0.00 | 0.28 | -0.20 | **0.66** | -0.01 | 0.55 | Friendliness + |
| Sociable | **0.57** | 0.04 | -0.15 | **0.64** | -0.04 | 0.75 | Friendliness + |
| Independent | 0.29 | 0.06 | **-0.50** | **-0.62** | 0.03 | 0.72 | Dominant + |
| Dependent | -0.32 | -0.06 | 0.34 | **0.59** | -0.08 | 0.58 | Confidence - |
| Solitary | **-0.47** | -0.26 | 0.01 | **-0.54** | 0.01 | 0.58 | Friendliness - |
| Friendly | 0.39 | **-0.45** | -0.27 | **0.51** | -0.08 | 0.69 | Friendliness + |

*(Continued)*

**Table 2.** (Continued)

| Item | Infant Macaque Components | | | | | | Corresponding Adult Component |
|------|--------|-------------|---------|-------------|-----------|-------|-------------------------------|
| | Openness | Assertiveness | Anxious* | Friendliness | Intellect* | $h^2$ | |
| Decisive | 0.02 | 0.19 | -0.22 | -0.19 | **0.73** | 0.65 | Friendliness + |
| Intelligent | 0.24 | 0.17 | -0.17 | -0.11 | **0.68** | 0.58 | Friendliness + |
| Disorganized | **0.46** | 0.00 | 0.08 | 0.12 | **-0.67** | 0.69 | Confidence - |
| Clumsy | 0.02 | 0.02 | -0.02 | 0.03 | **-0.59** | 0.35 | Activity - |
| Erratic | 0.20 | 0.33 | 0.19 | -0.21 | **-0.53** | 0.52 | Anxious + |
| Predictable | -0.23 | -0.34 | -0.17 | 0.16 | **0.45** | 0.43 | Activity - |
| Proportion of variance | 0.19 | 0.15 | -0.11 | 0.09 | 0.08 | | |

$N$ = 55. Salient loadings are in boldface.

*Indicates a component that has been reflected;

+ indicates a positive loading and

- indicates a negative loading with the corresponding adult personality component (if there is one).

N/A indicates the infant item has no corresponding adult structure component.

higher infant Assertiveness ($r$s = 0.97) in the infant four, five, and six component structures and higher infant Intellect ($r$ = 0.53) in the infant six component structure. Adult macaque Friendliness was significantly correlated with higher infant *Openness* ($r$s ≥ 0.52) and infant Friendliness ($r$s ≥ 0.80) and lower infant Anxiety ($r$s ≥ -0.59) in the four, five, and six component structures. Adult macaque Activity was significantly correlated with higher infant Openness ($r$s ≥ 0.90) and infant Assertiveness ($r$s = 0.44) in the infant four, five, and six component

**Table 3. Pearson correlations for infant personality components based on the six component adult personality structure (from [53]).**

| Infant Structure | Adult Structure | | | | | | | | | | | |
|------------------|-----------------|--|--|--|--|--|--|--|--|--|--|--|
| | Confidence | | Openness | | Assertiveness | | Friendliness | | Activity | | Anxiety | |
| **Four Component** | | | | | | | | | | | | |
| Openness | **0.67** | [0.50,0.80] | **0.95** | [0.91,0.97] | **0.52** | [0.30,0.69] | **0.52** | [0.30,0.69] | **0.90** | [0.83,0.94] | -0.33 | [-0.55,-0.08] |
| Assertiveness | **0.62** | [0.42,0.76] | 0.23 | [-0.04,0.46] | **0.97** | [0.94,0.98] | 0.15 | [-0.12,0.40] | **0.44** | [0.20,0.63] | -0.02 | [-0.29,0.24] |
| Anxiety | **-0.84** | [-0.90,-0.74] | -0.25 | [-0.48,0.02] | -0.30 | [-0.52,-0.04] | **-0.59** | [-0.74,-0.38] | -0.19 | [-0.44,0.08] | **0.88** | [0.80,0.93] |
| Friendliness | 0.02 | [-0.25,0.28] | 0.22 | [-0.04,0.46] | -0.21 | [-0.45,0.06] | **0.84** | [0.74,0.90] | 0.02 | [-0.24,0.29] | -0.32 | [-0.54,-0.06] |
| **Five Component** | | | | | | | | | | | | |
| Openness | **0.66** | [0.48,0.79] | **0.95** | [0.91,0.97] | **0.53** | [0.31,0.70] | **0.52** | [0.30,0.69] | **0.91** | [0.84,0.94] | -0.32 | [-0.54,-0.06] |
| Assertiveness | **0.62** | [0.42,0.76] | 0.23 | [-0.04,0.46] | **0.97** | [0.94,0.98] | 0.15 | [-0.12,0.40] | **0.44** | [0.20,0.63] | -0.02 | [-0.29,0.24] |
| Anxiety | **-0.88** | [-0.93,-0.80] | -0.42 | [-0.62,-0.17] | -0.33 | [-0.55,-0.07] | **-0.64** | [-0.78,-0.46] | -0.35 | [-0.56,-0.09] | **0.85** | [0.75,0.91] |
| Friendliness | 0.02 | [-0.25,0.28] | 0.22 | [-0.04,0.46] | -0.21 | [-0.45,0.06] | **0.84** | [0.74,0.90] | 0.02 | [-0.24,0.29] | -0.32 | [-0.54,-0.06] |
| Intellect | 0.13 | [-0.14,0.38] | **-0.50** | [-0.68,-0.27] | -0.10 | [-0.36,0.17] | 0.02 | [-0.25,0.28] | **-0.48** | [-0.66,-0.24] | **-0.45** | [-0.64,-0.21] |
| **Six Component** | | | | | | | | | | | | |
| Openness | **0.65** | [0.46,0.78] | **0.94** | [0.90,0.96] | **0.53** | [0.31,0.70] | **0.54** | [0.32,0.70] | **0.90** | [0.84,0.94] | -0.30 | [-0.53,-0.04] |
| Assertiveness | **0.62** | [0.42,0.76] | 0.23 | [-0.04,0.46] | **0.97** | [0.94,0.98] | 0.15 | [-0.12,0.40] | **0.44** | [0.20,0.63] | -0.02 | [-0.29,0.24] |
| Anxiety | **-0.88** | [-0.93,-0.80] | -0.42 | [-0.62,-0.17] | -0.33 | [-0.55,-0.07] | **-0.64** | [-0.78,-0.46] | -0.35 | [-0.56,-0.09] | **0.85** | [0.75,0.91] |
| Friendliness | -0.05 | [-0.32,0.21] | 0.17 | [-0.10,0.42] | -0.30 | [-0.52,-0.03] | **0.80** | [0.67,0.88] | -0.06 | [-0.32,0.21] | -0.30 | [-0.52,-0.04] |
| Predictability | -0.12 | [-0.38,0.15] | **-0.47** | [-0.65,-0.23] | -0.32 | [-0.54,-0.06] | 0.01 | [-0.25,0.28] | **-0.54** | [-0.71,-0.32] | -0.35 | [-0.56,-0.09] |
| Intellect | **0.72** | [0.57,0.83] | 0.33 | [0.06,0.54] | **0.53** | [0.31,0.70] | 0.24 | [-0.03,0.48] | 0.40 | [0.15,0.60] | **-0.50** | [-0.67,-0.27] |

Correlated are reported for infant macaque four, five, and six component personality structures. $N$ = 55. Boldface correlations are statistically significant ($p$s < 0.05), and 95% confidence intervals are in brackets.

**Table 4. Pearson correlation of subjective well-being and infant and adult personality structures.**

| Structure | SWB component | |
|---|---|---|
| | **SWB** | **95% CI** |
| **Infant** | | |
| **Four Component** | | |
| Openness | **0.72** | [0.56,0.83] |
| Assertiveness | **0.44** | [0.20,0.63] |
| Anxiety | **-0.74** | [-0.84,-0.58] |
| Friendliness | 0.33 | [0.08,0.55] |
| **Five Component** | | |
| Openness | **0.71** | [0.55,0.82] |
| Assertiveness | **0.44** | [0.20,0.63] |
| Anxiety | **-0.80** | [-0.88,-0.68] |
| Friendliness | 0.33 | [0.08,0.55] |
| Intellect | 0.05 | [-0.21,0.31] |
| **Six Component** | | |
| Openness | **0.71** | [0.54,0.82] |
| Assertiveness | **0.44** | [0.20,0.63] |
| Anxiety | **-0.80** | [-0.88,-0.68] |
| Friendliness | 0.26 | [-0.00,0.49] |
| Predictability | -0.01 | [-0.28,0.26] |
| Intellect | 0.50 | [0.27,0.68] |
| **Adult** | | |
| Confidence | **0.78** | [0.72,0.90] |
| Openness | **0.59** | [0.39,0.74] |
| Assertiveness | **0.45** | [0.22,0.65] |
| Friendliness | **0.69** | [0.50,0.80] |
| Activity | **0.60** | [0.39,0.74] |
| Anxiety | **-0.65** | [-0.80,-0.51] |

$N$ = 55. Boldface correlations are significant at $p < 0.05$; subjective well-being (SWB) 95% confidence intervals (CI) are reported in brackets.

structures and lower Intellect ($r$ = -0.48) in the infant five component structure and lower Predictability ($r$ = -0.54) in the infant six component structure. Adult macaque Anxiety was significantly correlated with higher infant Anxiety ($r$s ≥ 0.85) in the infant four, five, and six component structures, lower Intellect ($r$ ≥ -0.45) in the infant five and six component structures.

The subjective well-being component was significantly correlated with higher infant Openness ($r$s ≥ 0.71) and Assertiveness ($r$ = 0.44) and lower infant Anxiety ($r$s ≥ -0.74) in the infant macaque four, five, and six component structures, $p$s < .05 (Table 4). Infant subjective well-being also positively correlated with adult Confidence ($r$ = 0.78), Openness ($r$ = 0.59), Assertiveness ($r$ = 0.45), Friendliness ($r$ = 0.69), and Activity ($r$ = 0.60), and negatively correlated with adult Anxiety ($r$ = -0.65) on the adult macaque six component structure.

## Discussion

We tested whether adult-like personality factors are already present in infancy in rhesus macaque monkeys. We found infant macaques have a five component personality structure, based on caregiver ratings of 52 traits that observers showed agreement on: Openness (e.g.,

curiosity, inquisitive, playfulness), Assertiveness (e.g., dominance, bullying, aggressive), Anxiety (e.g., vigilance, fearful), Friendliness (e.g., sociable, affectionate, sympathetic), and Intellect (e.g., intelligent, decisive). These components are largely analogous to the six components in adult macaques—Openness, Assertiveness, Anxiety, Friendliness, Confidence, and Activity—although we also found some differences between adult and infant personality. These components in infant macaques are also similar to those reported in human children as young as 2 to 3 years of age, described as the "Little Six": Openness, Agreeableness, Neuroticism, Extraversion, Conscientiousness, and Activity [36,40,41]. Furthermore, we found links between personality and well-being: Infants' subjective well-being positively correlated with Openness and Assertiveness and negatively correlated with Anxiety, similar to findings in adult macaques [53]. These findings suggest stable structural personality components within this species.

## Interrater reliabilities of personality and subjective well-being

We found that all but two of the 54 HPQ items (unperceptive and imitative) and all four of the subjective well-being items were reliable among raters. Observers were not reliable on ratings of the item unperceptive, similar to results of observer ratings of adult macaques [53]. Previous studies reported that traits related to Extraversion have the highest levels of interrater agreement in both humans and animals, while traits related to Neuroticism have high levels of agreement in animals, but not humans, and traits related to agreeableness have the lowest levels of interrater agreement in both humans and animals [49]. For subjective well-being, our interrater reliabilities were also excellent, and comparable to those reported in adult macaques [53]. Together, these findings suggest that observers agreed on their ratings of infant macaque personality and well-being.

## Personality component structures in infant macaques

The five personality constructs we found in infant macaques—Openness, Assertiveness, Anxiety, Friendliness, and Intellect—appear similar to those reported in adult macaques [53], as well as other nonhuman primates [67,69,78,79,103] and human children [36,40,41]. Next, we outline each personality component that we detected in infant macaques and consider the similarities in these components with age and across species.

Infant macaques exhibit a component, which we call Openness, which may be similar to Surgency/Extraversion temperament structure reported in human infants and adult macaques, which refers to infants' tendency to exhibit energetic activity, positive affect, and high intensity pleasure [50], sometimes referred to as Surgency/Sociability (vs. Shyness/Inhibition) in children [41,104–106]. This component may be similar to the Openness component reported in 2- to 3-year-old children, which includes curiosity and exploring, love of learning, and interest in experiencing new things [36,40,41]. In adult macaques, higher levels of Openness are associated with better cognitive performance [107], so this may be an interesting personality component to study developmentally as it relates to learning.

Infant Assertiveness seems to mirror adult Assertiveness [53]. We decided to name this infant component Assertiveness, rather than Dominance, to avoid confusion with hierarchical dominance. Given the strict dominance hierarchy that rhesus macaques live in [108], it is unsurprising that traits relating to these behaviors would show up in infancy. Similar individual differences in aggressiveness have been reported in human infants and preschoolers, described as being low in Agreeableness, and high levels are sometimes described as having a "difficult temperament" (for a review, see [109]). In macaques, higher levels of assertiveness are associated with social success [110] and visible in facial morphology as a social signal [111], so it may be important to study this personality component in relation to the development of

social behaviors and skills. While the infant macaques in the present study were separated from their mothers, disrupting the usual rank inheritance transfer through social experiences [110,112], we still found this to be a distinct component, underscoring its potential importance.

Both infants and adults have a similar Anxiety component [53]. The Anxiety component we found here may be similar to the Negative Affectivity temperament structure reported in human and macaque newborns, which reflects an infant's tendency to experience negative emotions and distress [50]. Infant macaque Anxiety may also be similar to the Neuroticism component reported in human infants [36,40,41], and similar to Fearfulness in adult macaques [113]. In fact, in macaques, Fearfulness is reported to be one of the most stable personality traits across the first 7 years of life [113]. Higher levels of neuroticism are linked to a range of poor health outcomes, so a better understanding of its developmental origins is of significant clinical relevance (for a review, see [114]).

In macaques, both infants and adults have a component called Friendliness [53]. This personality component appears to be similar to the Sociability dimension reported in adult macaques, associated with being affiliative, warm, and less solitary [115], and the Extraversion component reported in human 2- to 3-year-olds [2,41,104–106]. Sociality is a core individual difference in primates, reported across a wide range of species [53,59,68]. Although we did not test for sex differences in Friendliness in the present study, sex differences in social behaviors are reported in both human and macaques, with females generally showing higher levels of social interest than males already in early infancy [88]. Understanding the causes and consequences of infants with low levels of sociability may help with the development of animal models to study developmental disorders, especially those that disproportionately affect males, such as autism [116,117].

We also found a dimension in infant macaques that was not apparent in adults: Intellect. The infant macaque Intellect component included items related to intelligence, such as being more thoughtful and more decisive, while being less distractible and less clumsy. In adult macaques, the items on this component load across Friendliness, Confidence, Activity, and Anxiety. Instead, this component appears to more closely resemble orangutan Intellect, with which it shares four of its six items [79]. Our findings suggest that this component—Intellect —may not be species-specific, found only in orangutans [80], but may be shared with other primates, at least at some points in development. This Intellect component also shares some similarities with the human toddler dimension Conscientiousness, which includes thoughtfulness, attentiveness, concentration, and planning [41]. Self-regulation, in particular, appears to be a core component of Conscientiousness in human children [118].

In the six component structure we found the items in the Intellect component from the five component structure were split into two components that were comprised of items relating to: (1) a Predictability component (predictable, not erratic, and not clumsy), and (2) an Intellect component (intelligent, organized). While we did not find Predictability in the five component structure, it did appear in the six component structure, suggesting that it may be an emerging component of personality, but may not be as stable as the other components at this age.

The current study offers novel insights into human infant personality and well-being and highlights future directions for research in human infants. Our findings in macaques suggest that there may already be well-established adult-like dimensions of personality detectable in infancy through caregiver report. To our knowledge, no studies, to date, have attempted to measure human adult personality (i.e., the "Big Five") in human infants, despite reports of similar dimensions (i.e., the "Little Six") in toddlers and young children (2 to 5 years of age) [36,40,41]. It may therefore be worthwhile to explore whether such personality components may be detected earlier in human infants as well. Such research may help to bridge the gap

between studies of infant temperament and studies of adult personality [41], which have historically relied on different instruments.

## Subjective well-being in infant macaques

We found that all four subjective well-being items loaded onto a single component, similar to adult macaques [53], orangutans [79], Western lowland gorillas [78], chimpanzees [80], and brown capuchins [76]. We also found that infant macaque subjective well-being positively correlated with Openness, Assertiveness (Dominance), and Friendliness, and negatively correlated with Anxiety. In adult macaques, higher confidence and friendliness, and lower anxiety are associated with higher subjective well-being [53], and similar patterns have been reported in chimpanzees [30], and humans [119]. In human adults, individuals who report higher levels of subjective well-being also tend to be lower in Neuroticism (associated with anxiety) and higher in Extraversion (associated with friendliness) [120]. These results suggest that well-being may be related to personality in similar ways across the lifespan and across primate species.

There are not yet well-established measures of human infant subjective-well being. The measure of subjective well-being used here may be adapted for use with human infants, and offers a number of advantages over previous measures. For example, in addition to including questions about positive emotions, this measure also includes questions about goal-achievement ("Estimate, for your infant, the extent to which he/she is effective or successful in achieving his/her goals or wishes"), infants' experiences of social interactions ("Estimate the extent to which social interactions with other people are satisfying, enjoyable experiences as opposed to being a source of fright, distress, frustration, or some other negative experience"), and asks raters to imagine themselves as the infant ("Imagine how happy you would be if you were your infant for a week. You would be exactly like your infant. You would behave the same way as your infant, would perceive the world the same way as your infant, and would feel things the same way as your infant."). This measure of well-being may be adapted for use with human infants and validated through comparisons with other similar measures, such as parent survey-based measures of infant positive affect [10, 11] and by comparing directly with infant behaviors (e.g., smiles, laughter, positive vocalizations). Validating a measure of subjective well-being in human infants could facilitate studies of the early developmental emergence of this construct, its stability across the lifespan, test potential associations between parent and infant well-being before and after birth, and associations between infant subjective well-being and personality. Recent studies on human infant subjective well-being report that parents' ratings of infant positive, but not negative affect, predict adult life satisfaction [10] as well as cognition in childhood and educational attainment in adulthood [121]. These studies suggest that subjective well-being in infancy may lay the foundation for later success in across numerous domains.

## Limitations and future directions

A limitation of the present study is that it included a relatively small sample of infant macaques reared in a neonatal nursery by humans. Future studies are needed to expand this research to larger and more diverse populations, including those socially reared in laboratories, zoos, field stations, the wild, and other contexts, to test the generalizability of these findings. At the same time, this nursery rearing is a strength of the current study because, despite being raised in nearly identical environments, we still found six different personality dimensions and variation among individuals in these dimensions. In addition, the infants in the current study grew up in very different environments relative to those of previous studies in wild populations

(Weiss et al., 2011), and yet our findings are largely similar. Together, these findings suggest that individual differences in these personality factors are unlikely to be exclusively due to variation in infants' postnatal environments, but rather, are more likely due (at least in part) to differences in infants' prenatal environment and/or genetics. The present study, therefore, offers fundamental insight about personality development, revealing its early ontogenetic roots.

The present study is also limited in that we only focused on one age group using a cross-sectional approach. Infant macaques at 7 months old are approximately equivalent to 2-year-old human infants, given that they are estimated to develop roughly four times faster than human infants, in their cognitive and brain development [122,123]. Given how little is know about infant personality development, both in human and nonhuman primates, future studies are needed in both younger and older infant macaques to determine how personality emerges. Studies in nonhuman primate infants will be instrumental in uncovering how infant temperament interacts with the early environment to shape development over time [124]. In addition, comparative studies exploring personality changes with age will be fruitful. For example, one study found chimpanzees and humans showed remarkable similarities in changes to their personality dimensions across the lifespan, from adolescence into adulthood [84]. Ideally, in future work, researchers could follow infants longitudinally and repeatedly sample their personality as they grow up, into adulthood, to better capture from birth through adulthood, across the lifespan, to further test the developmental stability of these dimensions [113]. A better understanding of infant primate personality may enable us to better identify infants who are outliers in specific traits associated with developmental disabilities (e.g., [125]), which may enable better animal models of human disabilities [117]. Through better understanding the development of personality, we may be able to enhance infants' development by pre-screening them to identify infants at risk of developing problems and helping them overcome temperament-related challenges, and by training caregivers to better align their responses to fit each child's characteristics [7,8,126].

## Conclusions

Our findings suggest that, in macaques, infant personality dimensions may be conceptually related to adult personality. Given that infant macaque are a popular model of human development, it is critical to understand the ways in which macaque personality may be similar to, or different from, that in humans. Further research is necessary to explore the antecedents, predictive validity, and stability of these personality components across situations and with development. Animal studies of personality can bring unique insights to the biological mechanisms that underlie personality development, including their causes and developmental plasticity. Considering that macaque infants are often studied as a model of human infant development, it is critical to understand the ways in which macaque infants may be similar to, or different from, human infants, in terms of personality. Nonhuman primate models of infant development offer unique insights about the development of personality and subjective well-being, widening our view of individual differences and their early emergence.

## Supporting information

**S1 File. Infant macaque code.** R code for statistical analysis that we used in the current study.
(R)

**S2 File. Infant macaque personality data.** Blinded raw data from the current study.
(XLSX)

**S1 Table. Promax rotated infant rhesus macaque four and six component models.** Salient loadings are in boldface. *N* = 55. *Indicates a component that has been reflected.
(XLSX)

**S2 Table. Promax-rotated component correlations of four, five, and six component structures.** Correlations from S1 Table promax-rotations.
(XLSX)

**S3 Table. Varimax rotated infant rhesus macaque four and six component models.** Salient loadings are in boldface. *N* = 55. *Indicates a component that has been reflected.
(XLSX)

**S4 Table. Factor analysis with varimax rotation of infant rhesus macaque structure.**
*N* = 55. Salient loadings are in boldface. *Indicates a component that has been reflected. Proportion of variance = 58%.
(XLSX)

**S5 Table. Interrater reliability of components derived from infant macaque five component structure.** Note. Based on 55 rhesus macaques. k = 2.8.
(XLSX)

**S6 Table. PCA of infant rhesus macaque subjective well-being items.** *N* = 55. Proportion of variance = 69%.
(XLSX)

## Acknowledgments

We thank Stephen J. Suomi and the animal care staff in the Laboratory of Comparative Ethology at the National Institutes of Health, USA. We thank Alexander Weiss for sharing his methodological and statistical expertise.

## Author Contributions

**Conceptualization:** Elizabeth A. Simpson.

**Data curation:** Elizabeth A. Simpson.

**Formal analysis:** Lauren M. Robinson.

**Methodology:** Elizabeth A. Simpson, Lauren M. Robinson, Annika Paukner.

**Project administration:** Elizabeth A. Simpson.

**Supervision:** Elizabeth A. Simpson.

**Writing – original draft:** Elizabeth A. Simpson, Lauren M. Robinson, Annika Paukner.

**Writing – review & editing:** Lauren M. Robinson, Annika Paukner.

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
