## [Decision Letter · Decision Letter 0]

15 Oct 2019

PONE-D-19-26868

Infant rhesus macaque (Macaca mulatta) personality and subjective well-being

PLOS ONE

Dear Dr. Simpson,

Thank you for submitting your manuscript to PLOS ONE. After careful consideration, we feel that it has merit but does not fully meet PLOS ONE’s publication criteria as it currently stands. Therefore, we invite you to submit a revised version of the manuscript that addresses the points raised during the review process.

There are several major issues you have to adress, principally:

- Data reduction used: PCA vs. FA

- Reliability of some of the items

- Discussion and the interpretation of the results

We would appreciate receiving your revised manuscript by Nov 29 2019 11:59PM. To enhance the reproducibility of your results, we recommend that if applicable you deposit your laboratory protocols in protocols.io, where a protocol can be assigned its own identifier (DOI) such that it can be cited independently in the future. For instructions see: http://journals.plos.org/plosone/s/submission-guidelines#loc-laboratory-protocols

We look forward to receiving your revised manuscript.

Kind regards,

Miquel Llorente, PhD

Academic Editor

PLOS ONE

Reviewers' comments:

Reviewer's Responses to Questions

**Comments to the Author**

1. Is the manuscript technically sound, and do the data support the conclusions?

Reviewer #1: No

Reviewer #2: Yes

2. Has the statistical analysis been performed appropriately and rigorously? 

Reviewer #1: No

Reviewer #2: Yes

3. Have the authors made all data underlying the findings in their manuscript fully available?

Reviewer #1: Yes

Reviewer #2: Yes

4. Is the manuscript presented in an intelligible fashion and written in standard English?

Reviewer #1: Yes

Reviewer #2: Yes

5. Review Comments to the Author

Reviewer #1: The overarching focus in this paper is in identifying the personality structure of a set of infant rhesus monkeys that were essentially nursery-reared – some had 2 hrs/day of peer socialization (while living alone) and the others lived in peer groups. Ratings were done when the animals were about 7 months of age. The authors then did a principal components analysis (PCA) to identify a structure to their data, and computed scores on 4, 5, and 6 dimensions. They also computed scores based on a published study of adult monkeys (Weiss et al., 2011), and compared the two sets of scores using Pearson product-moment correlations. The authors’ analysis suggests adult personality has definite antecedents in young animals. (There is a second instrument used in this study, one that assesses subjective well-being; this analysis does not really seem to belong in this paper, and could probably be eliminated.)

There are two major issues that I believe are problematic in this paper.

First, on line 357, the authors state: “We found that all but two of the 54 HPQ items (unperceptive and imitative) and all four of the subjective well-being items were reliable among raters.” Inspection of Table 1, however, indicates that several items on the personality inventory had extremely low reliabilities, the worst being Predictable, whose values are .01 and .04. A psychometrician would hardly call these items “reliable,” and if reliability is a criterion for inclusion in a PCA, then the authors need to justify their decision for including items whose ICC values are extremely low.

Second, the authors use the inappropriate principal components procedure to identify the personality dimensions. PCA is a data reduction technique; the more appropriate technique is factor analysis, which is aimed explicitly at identify the latent variables that explain the observed data. While I understand that there are many papers in the animal personality literature that have used PCA instead of FA, that does not make the practice acceptable. There are many resources in the literature and online describing the differences between PCA and FA, and which technique should be used under which circumstances. In this case, the goal of identifying personality factors (ie, latent traits) makes PCA the wrong technique. The authors might look at a paper by Costello and Osborne, who present a very accessible discussion of this and other relevant issues. (Costello, A.B. and J.W. Osborne. 2005. Best practices in exploratory factor analysis: Four recommendations for getting the most from your analysis. Practical Assessment, Research & Evaluation 10:1–9.)

Two more minor issues are:

References to the supplementary tables are incorrect. In the paragraph starting on line 270, the second line indicates Table S1 is the result of the promax rotation, but the next sentence indicates S1 contains the varimax rotation. Similarly, the correlations of the promax rotated factors is indicated as S3, but I find them in S2 instead. In general, the authors need to insure the references to the tables are correct and accurate.

In line 446ff, the authors suggest “that these six personality differences are unlikely to be due to infants’ postnatal environments, but rather, are more likely due to differences in infants’ prenatal environment and/or genetics.” It’s unclear what they authors are trying to say here. If what they mean is that their six factors were derived from animals with unusual rearing histories, and yet they seem to parallel the structure derived from adult rhesus living in very different circumstances (the Weiss et al paper), then this should be more clearly stated. If, however, what they are suggesting is that postnatal factors do indeed play little role in the development of personality, then I would consider that an overstatement (i.e. I would disagree with the statement “The present study, therefore, offers fundamental insight about personality development, revealing that it does not start after birth, but long before”). I simply don’t see that they have results that support that statement. In fact, they do have animals from two different rearing conditions – Do those two sets of animals differ from each other on these factors? If so, then presumably that is due to the postnatal environment. Moreover, earlier, they indicated their analysis suggested a five-factor solution. Why are they now interpreting the six-factor solution? Finally, their phrase “these six personality differences” seems inaccurate; they are not referring to actual differences, but rather individual difference *factors*.

Reviewer #2: This study tests an impressive number of infant monkeys to assess their personality, using the HPQ, and their wellbeing. However, the methodology suffers from a number of limitations, some of which the authors raise themselves in the Discussion. Given these limitations, these results must be taken with caution, especially their applicability to other populations or monkeys that have been mother reared in typical environments. Indeed, we know that atypical rearing can have long term consequences on the personality expression of primates (e.g., Freeman et al 2016 Developmental Psychobiology), as well as other long-term effects (e.g., Capitanio et al 2006 Nursey Rearing and Biobehavioral Organization), and I think this needs to be better recognized.

Additionally, this study is framed as being novel in testing the personality of young macaques. However, at the California Primate Research Center, such evaluations have been conducted for many years, testing thousands of young macaques (e.g., Capitanio, 2017, Variation in Biobehavioral Organization). This should be more clearly acknowledged.

Finally, and as I note below, it is proposed that macaques offer a good animal model for studying the development of personality. Given this, I would have appreciated greater consideration of how these results related to other aspects of the monkeys’ behavior (to provide a fuller perspective) and how these results relate to what is known about human personality development.

Here are my more specific comments are they arise throughout the article:

Given that your title refers to macaques, when I was reading the opening few lines of your paper it was unclear whether you were referencing literature on human or macaque/primate infants. I suggest you explicitly state that you are referring to human infant research. Furthermore, given this opening on human infant literature, I think your Discussion could also benefit from greater consideration about how these results relate to what we know for human infants and children and whether there are parallels between the species. Given that your proposal that macaques are a good model species for this ontogenetic research, it would be beneficial to hear your conclusions about that in the Discussion.

I do not think the subtitles in the introduction are needed, nor in the Discussion, unless this is a requirement of the journal, I would suggest you omit them.

Line 178 – In your methods, when describing the macaques’ rearing experience, I infer that for the first 5 weeks of their life the macaques are singly housed. Please state this explicitly for clarity.

Line 187 – how many raters total did you include?

Line 215 – why did you rate the monkeys at 7 months old? Is this related to a particular developmental milestone? Please provide more detail about the decision behind this sampling point. This information is key in better understanding the relevance and importance of your results when understanding the development of macaques. Therefore, I think more background information about the ontogeny of macaques in general, and how that maps onto human development, would be helpful given your framing that you are using macaques as a model species to understand human personality development.

Line 215 – please provide the range of the number of raters per subject as well as the mean. I see this is provided with Table 1, but I think it would be helpful to have it presented in the text too.

Do you have any physiological, cognitive or behavioral data that were collected at the time that you could use to further validate these personality measures, especially the measure of wellbeing? For example, were those rate with better wellbeing also those with better body condition, lower cortisol, more responsive to tests of cognition etc.? I think these kind of additional data would really strengthen your data and conclusions. Were those monkeys rated high on the intellect factor also those that performed better in concurrent (or later) cognitive tasks. Please include such meta data if you have it.

Is it possible to get ratings on these monkeys now that they are older to see how these infant ratings translated to when these monkeys were older (sensu Weinstein and Capitanio 2012 J Comp Psychol)?

Line 447 – “to more larger” sounds awkward, I suggest “to larger and more diverse”

Line 454 – how can you conclude “The present study, therefore, offers fundamental insight about personality development, revealing that it does not start after birth, but long before”? You tested monkeys seven months after birth so how do you know that personality development does not start at birth?

I found the raw data set an additional materials. However, I think it would be beneficial if you could also provide your R script along with your data set for full transparency of methods.

6. PLOS authors have the option to publish the peer review history of their article (what does this mean?). If published, this will include your full peer review and any attached files.

Reviewer #1: No

Reviewer #2: No

---

## [Author Response · Author response to Decision Letter 0]

23 Oct 2019

Reviewer #1: The overarching focus in this paper is in identifying the personality structure of a set of infant rhesus monkeys that were essentially nursery-reared – some had 2 hrs/day of peer socialization (while living alone) and the others lived in peer groups. Ratings were done when the animals were about 7 months of age. The authors then did a principal components analysis (PCA) to identify a structure to their data, and computed scores on 4, 5, and 6 dimensions. They also computed scores based on a published study of adult monkeys (Weiss et al., 2011), and compared the two sets of scores using Pearson product-moment correlations. The authors’ analysis suggests adult personality has definite antecedents in young animals. (There is a second instrument used in this study, one that assesses subjective well-being; this analysis does not really seem to belong in this paper, and could probably be eliminated.)

Response: Our analyses are not only focused on the existence of a personality structure in infant macaques, but also on how the personality constructs we discovered relate to subjective well-being. This additional analysis gives us predictive validity of the instrument as used with infant macaques as our findings mirror similar findings in adult macaques. Therefore, we chose to retain the subjective well-being component of this study. 

There are two major issues that I believe are problematic in this paper.

1. First, on line 357, the authors state: “We found that all but two of the 54 HPQ items (unperceptive and imitative) and all four of the subjective well-being items were reliable among raters.” Inspection of Table 1, however, indicates that several items on the personality inventory had extremely low reliabilities, the worst being Predictable, whose values are .01 and .04. A psychometrician would hardly call these items “reliable,” and if reliability is a criterion for inclusion in a PCA, then the authors need to justify their decision for including items whose ICC values are extremely low.

Response: We adjusted our wording throughout the manuscript to reflect observer agreement rather than reliability. It is a requirement that observers agree on their ratings, which they do.

 A psychometrician was consulted from design through to revision of this study. He said, “They're [the interrater reliabilities] ratios, so [low reliabilities] just means a lot of variance that's not true-score variance compared to the true score variance. That doesn't mean there's no true score variance, though, and you can tell because, if there was no true score variance, the items would not load on factors/components. It would also mean that, when getting ICCs of your scales (the factor scores), it would show no increase, but it does.” See Reise & Henson (2003) for a review of true score variance.

 Interrater reliabilities of scale items are typically in line with repeatabilities of behavior. For further information, see papers by Bell et al (2009) and McCrae and Mottus (2019).

Bell, A. M., Hankison, S. J., & Laskowski, K. L. (2009). The repeatability of behaviour: a meta-analysis. Animal Behaviour, 77(4), 771-783.

McCrae, R. R., & Mõttus, R. (2019). A new psychometrics: What personality scales measure, with implications for theory and assessment. Current Directions in Psychological Science.

Reise, S. P., & Henson, J. M. (2003). A discussion of modern versus traditional psychometrics as applied to personality assessment scales. Journal of Personality Assessment, 81(2), 93-103.

Velicer, W. F., & Jackson, D. N. (1990). Component analysis versus common factor analysis: Some issues in selecting an appropriate procedure. Multivariate Behavioral Research, 25(1), 1-28

2. Second, the authors use the inappropriate principal components procedure to identify the personality dimensions. PCA is a data reduction technique; the more appropriate technique is factor analysis, which is aimed explicitly at identify the latent variables that explain the observed data. While I understand that there are many papers in the animal personality literature that have used PCA instead of FA, that does not make the practice acceptable. There are many resources in the literature and online describing the differences between PCA and FA, and which technique should be used under which circumstances. In this case, the goal of identifying personality factors (ie, latent traits) makes PCA the wrong technique. The authors might look at a paper by Costello and Osborne, who present a very accessible discussion of this and other relevant issues. (Costello, A.B. and J.W. Osborne. 2005. Best practices in exploratory factor analysis: Four recommendations for getting the most from your analysis. Practical Assessment, Research & Evaluation 10:1–9.)

Response: Regarding the point about FA versus PCA, we are not in agreement with the reviewer here as the field has not reached agreement on the preferred method, likely because they both give extremely similar results. To quote Velicer and Jackson (1990), “Because factor score estimates are nearly identical to component scores, and are themselves determinate, it is difficult to understand how factor scores can be better and more generalizable.” FA and PCA have both been used in animal personality, even within the same paper. For example, Weiss et al. (2006) reported in Footnote 4 that they performed PCA and FA and found a virtually identical structure. 

 Given the reviewer’s comments, we ran a FA on the five factor structure and ran a test of congruence comparing it to the structure given using PCA and found the following congruences:

 RC1 RC2 RC3 RC4 RC5

MR1 1.00 0.32 0.39 0.20 0.34

MR2 0.32 1.00 0.19 -0.17 0.13

MR3 0.40 0.18 1.00 0.26 -0.20

MR4 0.21 -0.16 0.26 1.00 0.07

MR5 0.37 0.13 -0.21 0.08 1.00

 Given this and the fact that we’re comparing our structure with the published adult rhesus macaque structure, which was calculated using PCA, we have chosen to retain the PCA approach and have made the following addition to reflect this: “We also ran a factor analysis (see S4 Table) and compared the results to this structure using a congruence test. We found the results of both tests to be virtually identical (congruence = 1.00 across all corresponding components) and therefore continued with the PCA approach to be consistent with the method used in Weiss [53].” (Lines 352-355)

 Supplementary Table 4 includes the results of the FA.

Two more minor issues are:

3. References to the supplementary tables are incorrect. In the paragraph starting on line 270, the second line indicates Table S1 is the result of the promax rotation, but the next sentence indicates S1 contains the varimax rotation. Similarly, the correlations of the promax rotated factors is indicated as S3, but I find them in S2 instead. In general, the authors need to insure the references to the tables are correct and accurate.

Response: Thank you for bringing this to our attention. This has now been corrected in the manuscript. 

4. In line 446ff, the authors suggest “that these six personality differences are unlikely to be due to infants’ postnatal environments, but rather, are more likely due to differences in infants’ prenatal environment and/or genetics.” It’s unclear what they authors are trying to say here. If what they mean is that their six factors were derived from animals with unusual rearing histories, and yet they seem to parallel the structure derived from adult rhesus living in very different circumstances (the Weiss et al paper), then this should be more clearly stated. If, however, what they are suggesting is that postnatal factors do indeed play little role in the development of personality, then I would consider that an overstatement (i.e. I would disagree with the statement “The present study, therefore, offers fundamental insight about personality development, revealing that it does not start after birth, but long before”). I simply don’t see that they have results that support that statement. In fact, they do have animals from two different rearing conditions – Do those two sets of animals differ from each other on these factors? If so, then presumably that is due to the postnatal environment. Moreover, earlier, they indicated their analysis suggested a five-factor solution. Why are they now interpreting the six-factor solution? Finally, their phrase “these six personality differences” seems inaccurate; they are not referring to actual differences, but rather individual difference *factors*.

Response: We revised this section for clarity (Lines 584-593). Unfortunately, we do not have a large enough sample size in the current study to examine different rearing groups within our infant sample, but we agree that this is an interesting future direction.

1. Reviewer #2: This study tests an impressive number of infant monkeys to assess their personality, using the HPQ, and their wellbeing. However, the methodology suffers from a number of limitations, some of which the authors raise themselves in the Discussion. Given these limitations, these results must be taken with caution, especially their applicability to other populations or monkeys that have been mother reared in typical environments. Indeed, we know that atypical rearing can have long term consequences on the personality expression of primates (e.g., Freeman et al 2016 Developmental Psychobiology), as well as other long-term effects (e.g., Capitanio et al 2006 Nursey Rearing and Biobehavioral Organization), and I think this needs to be better recognized.

Response: Thank you for drawing our attention to these relevant publications, which we now reference in our introduction, emphasizing the value of animal models for enabling experimental manipulations to infants’ early rearing environments, while also acknowledging such variation is a limitation to generalizability of studies that focus only on one specific type of sample (nursery-reared infants), such as the present study (Lines 147-159). However, given that our findings in human-reared infant monkeys are largely similar to those reported in adult wild populations (Weiss et al., 2011), we think some claims of generalizability of these basic personality factors across different populations with various early environments are not unreasonable.

2. Additionally, this study is framed as being novel in testing the personality of young macaques. However, at the California Primate Research Center, such evaluations have been conducted for many years, testing thousands of young macaques (e.g., Capitanio, 2017, Variation in Biobehavioral Organization). This should be more clearly acknowledged.

Response: We added to our introduction a brief review of previous studies on infant macaque personality and clarify how the present study is novel in its approach to assessing personality exclusively using caretaker surveys developed for adult macaques to assess infants’ personality dimensions (Lines 145-169).

3. Finally, and as I note below, it is proposed that macaques offer a good animal model for studying the development of personality. Given this, I would have appreciated greater consideration of how these results related to other aspects of the monkeys’ behavior (to provide a fuller perspective) and how these results relate to what is known about human personality development.

Response: We agree that an important next step is to consider whether these personality dimensions are related to individual differences in behavior, as we mention in our discussion. Unfortunately, we do not have such data to include in the present study. We added discussion of how these results relate to what is known about human infant development throughout (e.g., Lines 497-499, 509-512, 519-527, 551-573).

Here are my more specific comments are they arise throughout the article:

4. Given that your title refers to macaques, when I was reading the opening few lines of your paper it was unclear whether you were referencing literature on human or macaque/primate infants. I suggest you explicitly state that you are referring to human infant research. Furthermore, given this opening on human infant literature, I think your Discussion could also benefit from greater consideration about how these results relate to what we know for human infants and children and whether there are parallels between the species. Given that your proposal that macaques are a good model species for this ontogenetic research, it would be beneficial to hear your conclusions about that in the Discussion.

Response: Thank you for pointing out these ambiguities. We added “human” and “nonhuman primate” to clarify the species throughout our introduction.

 We have also added to our Discussion, as suggested, additional links between the findings of the present study and the study of human infant personality (e.g., Lines 497-499, 509-512, 519-527, 551-573). 

5. I do not think the subtitles in the introduction are needed, nor in the Discussion, unless this is a requirement of the journal, I would suggest you omit them.

Response: Some of the subheadings have been removed from the Introduction and Discussion sections, as recommended. We retained some headings as required by the journal. 

6. Line 178 – In your methods, when describing the macaques’ rearing experience, I infer that for the first 5 weeks of their life the macaques are singly housed. Please state this explicitly for clarity.

Response: Yes, that is correct. We added that information to the “Subjects” section (Line 222).

7. Line 187 – how many raters total did you include?

Response: We clarify that we had six raters (Line 239).

8. Line 215 – why did you rate the monkeys at 7 months old? Is this related to a particular developmental milestone? Please provide more detail about the decision behind this sampling point. This information is key in better understanding the relevance and importance of your results when understanding the development of macaques. Therefore, I think more background information about the ontogeny of macaques in general, and how that maps onto human development, would be helpful given your framing that you are using macaques as a model species to understand human personality development.

Response: Thank you for pointing out this oversight. We added our justification for why we choose this age group (Lines 232-235) and also discuss what the “equivalent” approximate age is for human infants (Lines 595-597): “Infant macaques at 7 months old are approximately equivalent to 2-year-old human infants, given that they are estimated to develop roughly four times faster than human infants, in their cognitive and brain development [122,123].”

9. Line 215 – please provide the range of the number of raters per subject as well as the mean. I see this is provided with Table 1, but I think it would be helpful to have it presented in the text too.

Response: We now include this information (Line 250).

10. Do you have any physiological, cognitive or behavioral data that were collected at the time that you could use to further validate these personality measures, especially the measure of wellbeing? For example, were those rate with better wellbeing also those with better body condition, lower cortisol, more responsive to tests of cognition etc.? I think these kind of additional data would really strengthen your data and conclusions. Were those monkeys rated high on the intellect factor also those that performed better in concurrent (or later) cognitive tasks. Please include such meta data if you have it.

Response: No, unfortunately additional physiological, cognitive, or behavioral data on these infants are not available. We agree that these are important future directions.

11. Is it possible to get ratings on these monkeys now that they are older to see how these infant ratings translated to when these monkeys were older (sensu Weinstein and Capitanio 2012 J Comp Psychol)?

Response: No, unfortunately we no longer have access to these individuals.

12. Line 447 – “to more larger” sounds awkward, I suggest “to larger and more diverse”

Response: We corrected this typo.

13. Line 454 – how can you conclude “The present study, therefore, offers fundamental insight about personality development, revealing that it does not start after birth, but long before”? You tested monkeys seven months after birth so how do you know that personality development does not start at birth?

Response: We are not claiming that personality development does not start at birth. We reworded this section to avoid confusion (Lines 584-593): “Together, these findings suggest that individual differences in these personality factors are unlikely to be exclusively due to variation in infants’ postnatal environments, but rather, are more likely due (at least in part) to differences in infants’ prenatal environment and/or genetics. The present study, therefore, offers fundamental insight about personality development, revealing its early ontogenetic roots.”

14. I found the raw data set an additional materials. However, I think it would be beneficial if you could also provide your R script along with your data set for full transparency of methods.

Response: We have now included the R script in our supporting materials.

---

## [Decision Letter · Decision Letter 1]

12 Nov 2019

PONE-D-19-26868R1

Infant rhesus macaque (Macaca mulatta) personality and subjective well-being

PLOS ONE

Dear Dr. Simpson,

Thank you for submitting your manuscript to PLOS ONE. After careful consideration, we feel that it has merit but does not fully meet PLOS ONE’s publication criteria as it currently stands yet. Therefore, we invite you to submit a revised version of the manuscript that addresses the points raised during the review process, specially adressing the comments by reviewer 2. 

Although I personally consider that the manuscript improved considerably, issues regarding FA vs. PCA and reliability vs. agreement have to be adressed before our final acceptation.

We would appreciate receiving your revised manuscript by Dec 27 2019 11:59PM. To enhance the reproducibility of your results, we recommend that if applicable you deposit your laboratory protocols in protocols.io, where a protocol can be assigned its own identifier (DOI) such that it can be cited independently in the future. For instructions see: http://journals.plos.org/plosone/s/submission-guidelines#loc-laboratory-protocols

We look forward to receiving your revised manuscript.

Kind regards,

Miquel Llorente, PhD

Academic Editor

PLOS ONE

Reviewers' comments:

Reviewer's Responses to Questions

**Comments to the Author**

1. If the authors have adequately addressed your comments raised in a previous round of review and you feel that this manuscript is now acceptable for publication, you may indicate that here to bypass the “Comments to the Author” section, enter your conflict of interest statement in the “Confidential to Editor” section, and submit your "Accept" recommendation.

Reviewer #1: (No Response)

Reviewer #2: All comments have been addressed

2. Is the manuscript technically sound, and do the data support the conclusions?

Reviewer #1: No

Reviewer #2: Yes

3. Has the statistical analysis been performed appropriately and rigorously? 

Reviewer #1: No

Reviewer #2: Yes

4. Have the authors made all data underlying the findings in their manuscript fully available?

Reviewer #1: Yes

Reviewer #2: Yes

5. Is the manuscript presented in an intelligible fashion and written in standard English?

Reviewer #1: Yes

Reviewer #2: Yes

6. Review Comments to the Author

Reviewer #1: This is a revision of a paper focused on comparing scores on personality ratings of infant monkeys with scores using a component analysis conducted on adults of the same species, with a goal of seeing how the newly-derived infant components compare with the adult components. My previous comments focused on two main issues.

1. Reliability of individual items. The authors responded to my original concern that many items show low reliabilities in two ways.

First, the authors respond that they have altered the wording in multiple places to refer to these values as “agreement” and not “reliability.” This is incorrect. Agreement and reliability are different constructs, and in fact, the authors’ use of ICC(3,1) and ICC(3,k) reflect their focus on *consistency* in reliability, not agreement. Consider an example of three individuals rating five children on a trait. Person 1 gives ratings of 1, 2, 3, 4, 5. Person 2 gives ratings of 2, 3, 4, 5, 6, and Person 3 gives ratings of 3, 4, 5, 6, 7. Absolute agreement in these ratings = 0. The data are, however, quite consistent: for these data, ICC(3,1)=.714, and ICC(3,k)=.882. So to refer to the results of their reliability analysis as reflecting inter-rater agreement is incorrect.

Their second response seems to suggest that as long as an ICC is >0, there is some true score variance present, and they argue that the fact that these items loaded at all onto a factor is proof that, even with low reliabilities, they have value and so are retained. However, it’s generally considered that a measure is reliable if it *mostly* reflects true score variance, and not error variance. In the present case, 1% of the variance in Predictable is true score variance, and 99% is error. But “error” is not the same as “random error.” Using items with low true score variance can lead to spurious relationships owing to some other systematic source of variance embedded within the 99% error component of this item. Unfortunately, there are no hard and fast guidelines for where the line should be drawn so that items above the line are considered “reliable” and those below are considered “unreliable.” I have never seen that line at 1%, however, which is where the current authors are placing it. The authors will need to provide additional justification for their decision.

2. Principal Components Analysis (PCA) vs. Factor Analysis (FA). The idea that there are underlying latent traits that inform the display of behavior is fundamental to the idea of personality. Historically, the way these traits have been identified is through FA, although many in the ethological world use PCA instead. The underlying assumptions of these two procedures is different, as any text on factor analysis will indicate. The authors provide a reference by Velicer and Jackson (1990) as evidence that it really doesn’t make much difference in many cases between PCA and FA. However, the very next paper in that issue, by Gorsuch, does take this view to task. And contrary to the authors’ assertions, in human psychology anyway, FA does indeed seem to be the preferred method. The use of PCA in the animal literature is likely a hold-over historically (PCA is computationally easier than FA, but with the use of computers these days, this issue is moot) and disciplinary (PCA has its roots more in biology where data reduction – the principal reason for PCA – was the goal, not the discovery of latent traits). The authors did do a FA on their data, and a congruence analysis suggested the factor structure was virtually identical. This is reasonable; PCA and FA can, in many cases, lead to similar structures. The results of this analysis are presented in a Supplementary Table. Given that the present authors are trying to replicate a result from an earlier paper by Weiss that used PCA, provision of the FA results in a supplementary table is a sufficient response to my earlier critique.

Reviewer #2: Thank you for responding to all of my comments and suggestions and for providing greater theoretical context for your work in relation to human infant/child personality.

7. PLOS authors have the option to publish the peer review history of their article (what does this mean?). If published, this will include your full peer review and any attached files.

Reviewer #1: No

Reviewer #2: No

---

## [Author Response · Author response to Decision Letter 1]

21 Nov 2019

Reviewer #1

1. Reliability of individual items. The authors responded to my original concern that many items show low reliabilities in two ways.

First, the authors respond that they have altered the wording in multiple places to refer to these values as “agreement” and not “reliability.” This is incorrect. Agreement and reliability are different constructs, and in fact, the authors’ use of ICC(3,1) and ICC(3,k) reflect their focus on *consistency* in reliability, not agreement. Consider an example of three individuals rating five children on a trait. Person 1 gives ratings of 1, 2, 3, 4, 5. Person 2 gives ratings of 2, 3, 4, 5, 6, and Person 3 gives ratings of 3, 4, 5, 6, 7. Absolute agreement in these ratings = 0. The data are, however, quite consistent: for these data, ICC(3,1)=.714, and ICC(3,k)=.882. So to refer to the results of their reliability analysis as reflecting inter-rater agreement is incorrect.

1. Response: Given this feedback, we have adjusted the wording to interrater reliability, where appropriate, as previously written. 

2. Their second response seems to suggest that as long as an ICC is >0, there is some true score variance present, and they argue that the fact that these items loaded at all onto a factor is proof that, even with low reliabilities, they have value and so are retained. However, it’s generally considered that a measure is reliable if it *mostly* reflects true score variance, and not error variance. In the present case, 1% of the variance in Predictable is true score variance, and 99% is error. But “error” is not the same as “random error.” Using items with low true score variance can lead to spurious relationships owing to some other systematic source of variance embedded within the 99% error component of this item. Unfortunately, there are no hard and fast guidelines for where the line should be drawn so that items above the line are considered “reliable” and those below are considered “unreliable.” I have never seen that line at 1%, however, which is where the current authors are placing it. The authors will need to provide additional justification for their decision.

2. Response: We thank the reviewer for this thoughtful comment. The reviewer’s comment doesn’t take into account the fact that we aggregate all the ratings that reach 0.01 and above, which makes them more reliable and stable. Rushton, Brenerd, and Pressley speak to this exact point in their 1983 paper, when they state:

“Many important variables in behavioral development are presumed to be unrelated because of repeated failures to obtain substantial correlations. In this article, we explore the possibility that such null findings have often been due to failures to aggregate. The principle of aggregation states that the sum of a set of multiple measurements is a more stable and representative estimator than any single measurement. This greater representation occurs because there is inevitably some error associated with measurement. By combining numerous exemplars, such errors of measurement are averaged out, leaving a clearer view of underlying relationships.” 

 And also,

“According to the principle of aggregation, the sum of a set of multiple measurements is a more stable and unbiased estimator than any single measurement from the set. One reason is that there is always error associated with measurement. When several measurements are combined, these errors tend to average out, thereby providing a more accurate picture of relationships in the population. Perhaps the most familiar illustration of this effect is the rule in educational and personality testing that the reliability of an instrument increases as the number of items increases (e.g., Gulliksen, 1950; Lord & Novick, 1968).”

Regarding error variance they state,

“A more accurate picture is obtained by using the principle of aggregation and examining the predictability achieved from a number of measures. To reiterate, this effect occurs because the randomness in any one measure (error variance) is averaged out over several measures, leaving a clearer view of what a person's true behavior is like.”

Rushton, J. P., Brainerd, C. J., & Pressley, M. (1983). Behavioral development and construct validity: The principle of aggregation. Psychological Bulletin, 94(1), 18-38.

Regarding the reviewer never having seen the bar set to 0.01, we provide the following examples of papers where the limit is set to 0.01 and above:

Bergvall, U. A., Schäpers, A., Kjellander, P., & Weiss, A. (2011). Personality and foraging decisions in fallow deer, Dama dama. Animal Behaviour, 81(1), 101-112.

"The interrater reliabilities of 47 of the items were greater than 0 and used in the analysis.”

Lee, P. C., & Moss, C. J. (2012). Wild female African elephants (Loxodonta africana) exhibit personality traits of leadership and social integration. Journal of Comparative Psychology, 126(3), 224-232.

“The ICC (3, k) for two adjectives (apprehensive =-5.72 and tense =-2.57) were highly negative and so these were dropped from analyses. ICC for other adjectives ranged from 0.72 (slow, playful) to 0.02 (confident).”

Úbeda, Y., & Llorente, M. (2015). Personality in sanctuary-housed chimpanzees: A comparative approach of psychobiological and penta-factorial human models. Evolutionary Psychology, 13(1), 182-196. 

“The ICCs for the single (3, 1) and average (3, k) ratings were generally strong and there were no unreliable coefficients equal to or less than zero to eliminate from the analysis, indicating that raters tended to agree in their judgments about the personality traits of the chimpanzees.”

*The paper we used to make our adult personality comparisons: Weiss, A., Adams, M. J., Widdig, A., & Gerald, M. S. (2011). Rhesus macaques (Macaca mulatta) as living fossils of hominoid personality and subjective well-being. Journal of Comparative Psychology, 125(1), 72.

“We excluded the items autistic and unperceptive from further analysis as unreliable because their interrater reliabilities were less than zero.”

Although these studies include items above 0.0, one paper (Úbeda & Llorente, 2015) reported results that show two personality surveys produce similar results, another paper (Bergvall et al., 2011) showed that ratings correlated with behavioral tests and observations and foraging behavior (e.g., boldness predicted eating novel food), and a third paper (Weiss et al., 2011) demonstrated reliability across individuals and correlations with another measure, subjective well-being. Taken together, these findings suggest that the inclusion of aggregated items, even when reliability is low, do not impede the ability to predict other measures. Furthermore, all the items that we found were above 0.0, with the exception of autistic, were also reported to be reliable in the Weiss et al. study on the same species, with two primary differences being that they studied adult rhesus macaques and had more raters. This again shows that inclusion of these items is valid for comparison with this other study and that they have reliability across studies, even if they were low in ours. 

Finally, the interrater reliability of the components, including all items above 0.0, is as follows for the parallel analysis suggested five-component infant structure:

Item icc31 icc3k n_obs n_sub n_rat k

Openness5 0.6393677 0.8323313 154 55 6 2.8

Assertiveness5 0.6412943 0.8334954 154 55 6 2.8

Anxiety5 0.4812548 0.72204 154 55 6 2.8

Friendliness5 0.4920122 0.7305994 154 55 6 2.8

Intellect5 0.2081923 0.4240335 154 55 6 2.8

These results suggest that the inclusion of items 0.0 and above does not hamper the reliability at the component level. We’ve now included this table in the supplementary materials and referenced it in the paper (lines 277-279 and 321-323).

Given that there are no hard and fast guidelines, as this reviewer points out, and given that numerous previous studies in this area have used this approach, we therefore retained our interrater reliability analysis results.

3. Principal Components Analysis (PCA) vs. Factor Analysis (FA). The idea that there are underlying latent traits that inform the display of behavior is fundamental to the idea of personality. Historically, the way these traits have been identified is through FA, although many in the ethological world use PCA instead. The underlying assumptions of these two procedures is different, as any text on factor analysis will indicate. The authors provide a reference by Velicer and Jackson (1990) as evidence that it really doesn’t make much difference in many cases between PCA and FA. However, the very next paper in that issue, by Gorsuch, does take this view to task. And contrary to the authors’ assertions, in human psychology anyway, FA does indeed seem to be the preferred method. The use of PCA in the animal literature is likely a hold-over historically (PCA is computationally easier than FA, but with the use of computers these days, this issue is moot) and disciplinary (PCA has its roots more in biology where data reduction – the principal reason for PCA – was the goal, not the discovery of latent traits). The authors did do a FA on their data, and a congruence analysis suggested the factor structure was virtually identical. This is reasonable; PCA and FA can, in many cases, lead to similar structures. The results of this analysis are presented in a Supplementary Table. Given that the present authors are trying to replicate a result from an earlier paper by Weiss that used PCA, provision of the FA results in a supplementary table is a sufficient response to my earlier critique.

3. Response: We are pleased that the reviewer is happy with our response and accepts our use of PCA in this paper.

Regarding latent variables, Velicer and Jackson also respond to this in their 1990 paper by stating, “If the latent variable approach results in greater generalization under sampling, the factor analyses on the subsample should be closer to the population than the component analysis. Velicer (1972), using several existing well known data sets, found no observable difference.” and “Latent variable procedures (i.e., factor analysis) are asserted to have greater generalizability to the set of unsampled variables and thus represent an advantage for factor analysis. However, empirical studies (Velicer, 1974; Velicer & Fava, 1987, 1990) have generally found no difference between the methods under conditions of variable sampling. This issue does not represent a basis for selecting either method of analysis.”

Reviewer #2

1. Thank you for responding to all of my comments and suggestions and for providing greater theoretical context for your work in relation to human infant/child personality.

1. Response: We thank this reviewer for the positive comments and are happy that the reviewer is satisfied with our changes.

---

## [Decision Letter · Decision Letter 2]

6 Dec 2019

Infant rhesus macaque (Macaca mulatta) personality and subjective well-being

PONE-D-19-26868R2

Dear Dr. Simpson,

We are pleased to inform you that your manuscript has been judged scientifically suitable for publication and will be formally accepted for publication once it complies with all outstanding technical requirements.

With kind regards,

Miquel Llorente, PhD

Academic Editor

PLOS ONE

Additional Editor Comments (optional):

Reviewers' comments:

Reviewer's Responses to Questions

**Comments to the Author**

1. If the authors have adequately addressed your comments raised in a previous round of review and you feel that this manuscript is now acceptable for publication, you may indicate that here to bypass the “Comments to the Author” section, enter your conflict of interest statement in the “Confidential to Editor” section, and submit your "Accept" recommendation.

Reviewer #1: All comments have been addressed

2. Is the manuscript technically sound, and do the data support the conclusions?

Reviewer #1: (No Response)

3. Has the statistical analysis been performed appropriately and rigorously? 

Reviewer #1: (No Response)

4. Have the authors made all data underlying the findings in their manuscript fully available?

Reviewer #1: (No Response)

5. Is the manuscript presented in an intelligible fashion and written in standard English?

Reviewer #1: (No Response)

6. Review Comments to the Author

Reviewer #1: The remaining issue pertained to inclusion of individual items with low inter-rater reliability in the authors’ PCA. The authors’ response, namely to invoke the Principle of Aggregation, really is tangential to the issue – yes, aggregating individual items will result in a more reliable measure. The issue is *which* items to aggregate. The authors agree there are no hard and fast rules. I will simply note that if one contrasts the ICC values for the personality items in Table 1, comparing the values for those items that loaded in their PCA (n=45) with those items that did not (n=3: conventional, sensitive, quitting), that significant differences (both p<.01) exist for both ICC items. This suggests, at least to me, that individual level reliabilities *do* matter when selecting measures for PCA/FA.

7. PLOS authors have the option to publish the peer review history of their article (what does this mean?). If published, this will include your full peer review and any attached files.

Reviewer #1: No

---

## [Editor Report · Acceptance letter]

10 Dec 2019

PONE-D-19-26868R2 

Infant rhesus macaque (Macaca mulatta) personality and subjective well-being 

Dear Dr. Simpson:

I am pleased to inform you that your manuscript has been deemed suitable for publication in PLOS ONE. Congratulations! Your manuscript is now with our production department. 

With kind regards,

on behalf of

Dr. Miquel Llorente 

Academic Editor

PLOS ONE